# The Effect of Barley and Lysine Supplementation of Pasture-Based Diet on Growth, Carcass Composition and Physical Quality Attributes of Meat from Farmed Fallow Deer (*Dama dama*)

**DOI:** 10.3390/ani9020033

**Published:** 2019-01-24

**Authors:** Eva Kudrnáčová, Daniel Bureš, Luděk Bartoň, Radim Kotrba, Francisco Ceacero, Louwrens C. Hoffman, Lenka Kouřimská

**Affiliations:** 1Institute of Animal Science, Přátelství 815, Prague 22 – Uhříněves 104 00, Czech Republic; bures.daniel@vuzv.cz (D.B.); barton.ludek@vuzv.cz (L.B.); maugli46@volny.cz (R.K.); 2Department of Food Science, Faculty of Agrobiology, Food and Natural Resources, Czech University of Life Sciences Prague, Praha – Suchdol 165 21, Czech Republic; kourimska@af.czu.cz; 3Department of Animal Science and Food Processing, Faculty of Tropical AgriSciences, Czech University of Life Sciences Prague, Praha – Suchdol 165 21, Czech Republic; ceacero@ftz.czu.cz; 4Centre for Nutrition and Food Sciences, Queensland Alliance for Agriculture and Food Innovation, University of Queensland, 306 Carmody Road, St. Lucia, QLD 4069, Australia; louwrens.hoffman@uq.edu.au; 5Department of Animal Sciences, University of Stellenbosch, Private Bag XI, Matieland, Stellenbosch 7602, South Africa; 6Department of Microbiology, Nutrition and Dietetics, Faculty of Agrobiology, Food and Natural Resources, Czech University of Life Sciences Prague, Praha – Suchdol 165 21, Czech Republic

**Keywords:** diet, meat quality, muscle, performance, venison

## Abstract

**Simple Summary:**

Interest in deer farming has become more popular in European countries and, thus, production of venison under farm conditions is of utmost importance. Despite an increasing demand and interest in venison, only a few studies summarizing the effects of different diets on the growth, carcass parameters and the quality of meat from fallow deer have been published. Our study was the first to evaluate the effects of dietary barley and lysine supplementation on carcass parameters and physical quality of venison from farm-raised fallow deer. Supplementation with barley increased live weight gain, slaughter and carcass weights and fatness characteristics, whereas the addition of lysine to the barley-supplemented diet reduced the amounts of internal and carcass fat.

**Abstract:**

Fallow deer (*Dama dama*) are important meat producing species providing venison and other products to an international market. The present study investigated the effects of different feed rations on the growth, carcass characteristics and physical attributes of the *longissimus lumborum* (LL) and *semitendinosus* (SET) muscles of 45 farm-raised male fallow deer. The animals were divided into three separate groups: 15 pasture-fed (P), 15 pasture-fed and supplemented with barley (B), and 15 pasture-fed and supplemented with barley and lysine (BL). The animals were slaughtered at an average age of 17 months at three time points: after 155, 169 and 183 days on feed. The addition of barley to the feed ration significantly increased weight gain and had positive effects on slaughter and carcass weights, dressing-out proportion, carcass composition, the weight of LL muscle, and increased the redness, yellowness and chroma values of LL muscle. The supplementation with lysine reduced the amounts of carcass and internal fats without compromising other economically important traits.

## 1. Introduction

The continuous growth of the global population has been paralleled by increased demand for animal-based proteins [1]. During the last 50 years, average per capita meat consumption has increased by more than 45% [2]. Under farm conditions, meat can be produced from a variety of unconventional livestock species [3], among which, the utilization of farmed deer species and the production of deer meat (venison) are of particular interest and importance [4,5,6].

Deer carcass characteristics determine deer meat quantity and are associated with the market value of the deer [7]. These characteristics are largely dependent on the animal’s diet, which affects not only carcass composition but also the physical, chemical, technological and organoleptic properties of the meat [8]. The feeding regime and type of feed consumed by animals prior to slaughter alter the flavor and composition of the meat [9]. Pasture is a common but limited source of feed for farmed deer species, with simple grain-based diet supplementation essential, especially during the dry or wet and muddy season or to improve animal performance [10]. In general, pasture fattening is responsible for lean meat, whereas animals supplemented with concentrates, such as grain, tend to have a higher dressing-out proportion, a higher proportion of individual carcass parts, more intramuscular fat (IMF) and carcass fat [9,11,12,13].

A variety of diet-based methods for increasing meat production have been evaluated. Meat production generally increases with increasing dietary crude protein levels [14]. Previous studies dealing with different diets [11,12,13,15,16,17] have revealed differences in growth and carcass composition between deer grazing pasture and those supplemented with grain-based diets. In general, deer fed/supplemented with concentrates had higher carcass weights and dressing-out proportions than animals grazing pasture. Moreover, feeding concentrates may also affect the proportions of anatomical joints and lean cuts, as well as the amount of separable fat, as reviewed by Kudrnáčová et al. [18].

Only a few studies of the effects of grain-based supplementation on technological meat quality parameters have been carried out in cervids. Among studies of the effects of different diets on the physical attributes (pH, color, and shear force) of deer meat, Volpelli et al. [10] found no significant effects of different dietary treatments, whereas others observed significantly lower pH values and more tender meat in deer supplemented with concentrates compared with animals grazing on pasture [9,17,19].

Carcass conformation, meat quality and quantity depend on the amount of muscle and skeletal tissue. The amounts of theses tissues are related to the accretion of muscle proteins whose deposition requires a dietary supply of amino acids (AAs). Lysine is an essential AA that serves as a building block for protein biosynthesis [20]. Previous research has mostly focused on the importance of lysine in the diets of monogastric animals, such as pigs or poultry [20,21,22,23]. However, since the identification of lysine as one of the top limiting AAs for ruminants, research has expanded to include polygastric animals [14,24,25,26] and, very recently, cervids [27]. Lysine seems to be a limiting AA in the ruminal synthesis of microbial protein [28,29]. Thus, ruminants have higher metabolizable protein requirements that may exceed their supply [14,30]. Microbial protein breakdown and synthesis result in alterations of AAs from the animal’s diet upon absorption in the gut [31]. It is therefore essential to provide these AAs in a resistant/protected form that can withstand microbial degradation in the rumen and ensure availability in the abomasum. These preparations, referred to as rumen-protected amino acids (RPAAs), can be efficiently utilized by ruminants and thus potentially improve growth performance and carcass composition [32].

Venison is often considered an “organic” and safe product, with health-promoting characteristics that make it attractive to consumers and support its place in the human diet [5,33]. Growth, carcass composition and meat quality are of the utmost importance when the economic value of an animal is evaluated. Worldwide, the fallow deer (*Dama dama*) (FD) is one of the most abundant deer species raised under farm conditions, but only a few studies have focused on dietary concentrate and AA supplementation in this species. Despite extensive efforts to measure and quantify the carcass traits and physical attributes of deer meat, only a few investigations have targeted FD, and therefore considerable information gaps remain. Moreover, there have been no studies of the potential effects of AA supplementation on carcass parameters or technological meat quality parameters. The present study therefore sought to compare the growth, carcass composition and physical quality traits of venison from farmed FD males fed different diets.

## 2. Materials and Methods

### 2.1. Animals, Experimental Design, and Diets

All experimental procedures were approved by the Animal Care Committee of the Institute of Animal Science (IAS; IACUC No. 60444/2011-MZE-17214). A total of 45 FD bucks at an initial age of 11 months and an average live weight of 28.2 ± 1.8 kg were utilized. The animals were fattened during 2015 on the Mnich farm near Kardašova Řečice situated in the South Bohemian Region (49°16′71.9″ N; 14°90′05.2″ E). All animals originated from the same herd. Individual bucks were identified with plastic ear tags and, based on their body weights, were assigned to three separate groups of 15 animals housed in three 2-ha adjoining paddocks. The bucks were weighed three times (until the onset of fraying) during the experiment, and paddocks were switched among the groups at six-week intervals.

The groups were assigned to three different dietary treatments. Group P received only pasture, Group B received pasture supplemented with barley, and Group BL received pasture supplemented with barley mixed with lysine (LysiPEARL^TM^ at 5 g/day). The LysiPEARL^TM^ preparation (Kemin Industries, Inc., Des Moines, IA, USA) consisted of 50% of synthetic lysine and 50% hydrolyzed palm oil and provided lysine in encapsulated form (RPAA). The encapsulation protected the lysine from rumen microorganisms and ensured its release in the abomasum. Supplementation was performed once daily via wooden troughs placed in the pasture. One meter of trough length was available for each animal. All groups received a mineral mixture lick (Premin Slanisko, VVS Verměřovice Ltd., Verměřovice, Czech Republic). The finishing period was divided into two phases. In the first phase (90 days from the end of April until the end of July; Summer), the B and BL groups received barley in the amount of 0.2 kg/day/animal, whereas, in the second phase (on average 79 days, from the beginning of August until slaughter in October; Autumn), the dose of barley was increased to 0.4 kg/day/animal. Group BL received the same amount of lysine (5 g/day/animal) over the entire experiment.

### 2.2. Feed Chemical Composition

The average chemical compositions of the barley and pasture are shown in Table 1. During the experiment, forage samples were collected three times from three locations within each paddock (end of April, beginning of August, and October). All diet samples were freeze-dried (Freeze dryer ALPHA 1–4 LSC, Martin Christ Gefriertrocknungsanlagen GmbH, Osterode am Harz, Germany), and the average nutrient composition was analyzed as described by Jančík et al. [34]. The chemical composition of pasture was determined according to the following methods: dry matter: oven drying for 6 h at 105 °C to a constant weight; ash: oven drying for 6 hat 550 °C; crude fat: extraction for 6 h with petroleum-ether using Soxtec 1043 (FOSS Tecator AB, Höganäs, Sweden); nitrogen: Kjeldahl method (Kjeltec AUTO 1030 Analyser, Höganäs, Sweden) according to AOAC 976.05 [35]; crude protein: calculated as N × 6.25; acid detergent fiber (ADF) and lignin: determined according to AOAC 973.18 [35]; and neutral detergent fiber (NDF): analyzed in the presence of sodium sulfite and with α-amylase [36].

### 2.3. Slaughter Processing, Carcass Composition and Muscle Sampling

The experiment was terminated in October with the slaughtering of the bucks at an average age of 17 months. On each of three slaughter days (155, 169 and 183 days on feed, respectively), 15 animals (5 from each group) were randomly selected and stunned with a captive bolt within a handling box, weighed (slaughter weight—used for the calculation of slaughter characteristics), bled and eviscerated directly on the farm, and then transferred in a refrigerator truck to the experimental slaughterhouse of IAS for further processing. The weights of internal fat depots (the sum of the kidney, rumen and scrotal fat) were recorded. Within 5 h after slaughter, the carcasses were uniformly dressed and divided into two halves, and the carcass weights were taken. The dressing-out proportion was calculated as: (carcass weight/slaughter weight) × 100.

After chilling for 96 h, the cold carcass weights were recorded, and the right sides were divided into standardized commercial joints. The joints were separated into lean meat, bones, tendons, and separable fat (subcutaneous and intermuscular), and their respective weights were recorded. The total meat yield was calculated as the lean meat from all joints plus the lean trimmings. The meat from the trimmed rump, shoulder, loin and tenderloin was considered high-priced meat, and the lean meat from the remaining joints plus the lean trimmings were considered low-priced meat. The whole *longissimus lumborum* (LL) and *semitendinosus* (SET) muscles were collected from the right side and transported in a cooling box to the laboratory for further analyses.

### 2.4. Physical Analysis

pH readings were obtained using a puncture probe (SenTix Sp) connected to a pH 3310 m (WTW, Weilheim, Germany) at 96 h *post-mortem* in the LL and SET samples these pH_96h_ readings were considered the ultimate pH values (pH_u_). Instrumental color was measured on LL and SET samples 96 h *post-mortem* using a portable spectrophotometer (CM-2500d, Minolta, Osaka, Japan; aperture size of 8 mm including the specular component and 0% UV; illuminant/observer of D65/10°; zero and white calibration). The results were expressed by the *L** (lightness), *a** (redness) and *b** (yellowness) co-ordinates of the CIELab colorimetric space [37]. Three measurements per sample distributed on the muscle cross-section surface were taken after 30 min of air exposure to allow blooming, and an effort was made by the operator to avoid areas of dense connective tissue or fat. The *a** and *b** values were subsequently used to calculate chroma (*C**), saturation index = (*a**^2^ + *b**^2^)^0.5^ and hue angle (°) = tan^−1^ (*b**/*a**). The mean values of the three measurements of each attribute were determined for each muscle from each animal, and these values were used for statistical analyses.

Shear force was measured in cooked samples of LL and SET muscles that had been previously aged for 14 days (within whole carcass halves for the first four days and then vacuum-packed in plastic bags and aged at +4 °C for an additional 10 days). The samples were removed from the vacuum packaging and cut into 20-mm thick-steaks. The steaks were cooked on a double-glass/ceramic-plate grill (VCR 6l TL, Fiamma, Aveiro, Portugal) preheated to 200 °C until an internal temperature of 70 °C was reached, as determined by a digital temperature probe (AD14TH, Ama-Digit, Kreuzwerheim, Germany). The cooked samples were subsequently cooled to 4 °C and the center of each steak was divided into four rectangular blocks (20 mm × 10 mm × 10 mm) by cutting perpendicular to the muscle fiber direction. Care was taken to ensure that no visible connective tissue was included in the core. The peak force required to shear the samples across the fibers was recorded using an Instron Universal Texture Analyzer 3365 (Canton, MA, USA) fitted with a V-shaped Warner–Bratzler (WB) shear blade running at a crosshead speed of 100 mm/min. The maximum mean force in newtons (N) required to shear through the sample was based on at least nine measurements for each muscle from each animal.

### 2.5. Statistical Analysis

All calculations were performed in the statistical package SAS [38]. Each variable was previously tested for normality using the Kolmogorov–Smirnov goodness-of-fit test and for homogeneity of variance with the Levene test. Data were analyzed with a mixed linear model, and parameters were estimated by the restricted maximum likelihood method of the MIXED procedure. The statistical model involved the fixed effect of diet and the random effect of day of slaughter. The data in tables are presented as least squares means (LSM) and standard errors of the mean (SEM; *n* = 15). For *post hoc* analysis, Tukey’s range tests were used. Differences were considered significant at the level of *P* < 0.05.

## 3. Results

### 3.1. Animal Performance

The growth performance results and slaughter traits over the entire experiment are presented in Table 2. The average live weight of animals at the beginning of the experiment was 28.2 ± 1.8 kg, and the average slaughter weight was 48.5 ± 3.5 kg. As expected, there were no differences in initial weight (*P* > 0.05) among the three dietary treatment groups. Significant differences attributable to the different dietary treatments were observed in the daily weight gain during both fattening phases. The live weight at the end of Fattening Phase I tended to be higher in Groups B and BL than in Group P (*P* = 0.054). Groups B and BL had faster daily gain than Group P in both Fattening Phase I (*P* < 0.062), Fattening Phase II (*P* < 0.001), and over the entire experiment (*P* < 0.001). The differences in gain between the grain-supplemented Groups B and BL and the grazing Group P were 30.3 and 29.4 g/day, respectively. Moreover, at slaughter, bucks from Groups B and BL were 5.2 and 4.5 kg heavier on average, respectively, than Group P animals (*P* < 0.001). The grain-fed deer (Groups B and BL) produced carcasses that were 5.4 and 4.8 kg heavier, respectively, with higher dressing-out proportions (*P* < 0.001) than the pasture-fed (Group P) animals. However, the addition of lysine to the barley had no significant effect on any of the growth or carcass dress-out proportions when compared with barley supplementation alone. Barley supplementation resulted in a marked increase in the proportion of internal fat, which was nearly 2.5-fold higher in Group B than in Group P (*P* < 0.001). Group BL had a significantly lower proportion of internal fat (*P* < 0.05) than Group B but a significantly (*P* < 0.001) higher proportion than Group P.

### 3.2. Carcass Composition

The different dietary treatments significantly affected most of the carcass composition traits (Table 3). Group P produced leaner and lighter carcasses with lower amounts of meat, bones and tendons, and separable fat (*P* < 0.001) compared with Groups B and BL. In addition, the amount of separable fat was higher in Group B carcasses than in Group BL carcasses (*P* < 0.001).

Group P had the lowest amount of high-priced meat in the carcass (*P* < 0.001), but the proportion of the right-side weight in the Group P was higher (*P* < 0.001) than that in Group B and similar to that in Group BL. With respect to high-priced meat, Group P had the highest proportion of meat from the rump (*P* < 0.001), but Group BL had the highest proportion of meat from the shoulder (*P* < 0.001) and a higher proportion of meat from the tenderloin than Group B (*P* < 0.05). Compared with Groups B and BL, pasture-fed animals (Group P) exhibited a lower proportion of low-priced meat (*P* < 0.001) and a lower ratio of meat to bones (*P* < 0.001), but a higher proportion of bones and tendons and ratio of high- to low-priced meat (*P* < 0.001 and *P* < 0.01, respectively).

### 3.3. Physical Quality Attributes

The weights, pH values, Warner–Bratzler shear force and color parameters measured in LL and SET muscles of FD bucks are presented in Table 4. None of the physical components differed significantly except where stated. The supplemented diets resulted in heavier LL muscles in Groups B and BL compared with Group P (*P* < 0.05), whereas no apparent effect of diet (*P* = 0.472) on the weight of the SET muscle was observed among the individual groups. In addition, diet had no apparent effect on the pH_u_ values of either LL or SET muscles after slaughter.

Although the differences in most color parameters of the muscles were not significant, the tendency in color was similar. Diet appeared to have a significant effect on the redness (*a**) and yellowness (*b**) parameters of the LL muscle, with more intensely red and yellow meat in Group B than in Group P (*P* < 0.001 and *P* < 0.01, respectively). The highest values of saturation index (chroma) were observed in the LL muscle from Group B (*P* < 0.001).

## 4. Discussion

This experiment in FD bucks was performed with the aim of evaluating the effect of diet on the growth rates, carcass composition, and physical quality attributes as measured in two muscles. The bucks were reared under identical conditions, and all entered the experiment with similar average live weights. The animals were assigned to three different dietary treatment groups: P (pasture fattening), B (pasture fattening + barley supplementation) and BL (pasture fattening + supplementation with barley and RPAA lysine). Lysine was added as this essential amino acid is frequently insufficient in feed rations for ruminants [27,28]. To our knowledge, this study was the first to focus on the effects of RPAA lysine on the growth, carcass characteristics and physical quality attributes of FD.

### 4.1. Growth Performance

Barley-supplemented animals clearly showed greater daily live weight gain during the entire experimental period, in agreement with previous reports of positive effects of concentrates on growth performance compared with pasture-fed deer [15,18]. Diet was a significant source of variation of daily weight gain, particularly during the second fattening phase, during which Groups B and BL received a double dose of barley (0.4 kg/day per animal) and extreme drought conditions during the summer months resulted in inadequate pasture capacity. During this period, Groups B and BL had greater weight gain than the pasture-fed deer (Group P). Bovolenta et al. [39] also reported higher growth rates for FD supplemented with concentrates compared with those only grazing pasture. This difference was associated with the poor herbage allowance, which was not able to fully satisfy the nutritional requirements of the pasture-fed animals.

The differences between Groups B and BL and thus the effect of RPAA lysine on growth performance were smaller in magnitude. These results are broadly in agreement with those of Huang et al. [27], who reported no effect of lysine supplementation (crude protein-deficient diets with 3 g/kg lysine, 3 g/kg lysine + 1 g/kg methionine, and 3 g/kg lysine + 2 g/kg methionine) on growth performance in sika deer. Similarly, Hussein and Berger [24] and Klemesrud et al. [29] found no significant differences in performance responses to supplemental RPAA lysine in cattle.

### 4.2. Slaughter Traits

In this study, concentrate supplementation of FD clearly increased carcass weights, similar to other studies of concentrate feeding of FD [11,17], red deer (RD) [12] and reindeer [16]. By contrast, Wiklund et al. [13] reported slightly higher carcass weights of RD grazing natural pasture compared with those fed a pelleted feed mixture, although this difference was not significant.

In the present study, good dressing-out proportion values were observed that were similar to those reported previously for FD [11,39,40,41] but lower than those observed by Wiklund et al. [17]. In agreement with other studies comparing carcass traits in deer [11,12], the supplemented deer from Groups B and BL produced heavier carcasses and had higher dressing-out proportions. Similar effects of diet on internal fat deposition, as demonstrated in the present study, were reported by Phillip et al. [12] in RD fed diets with different ratios of concentrate to dried and pelleted roughage.

Notably, supplementation of FD with lysine (Group BL) significantly decreased internal fat and carcass fat proportions in this study. Lipogenesis inhibition and repartitioning of nutrients towards lean tissue deposition rather than fat deposition as a result of increasing dietary lysine levels have been reported in broilers [21,42,43], ducks [44] and pigs [23]. However, the underlying cause of the lower internal fat proportion in FD supplemented with lysine is not clear and requires further investigation.

### 4.3. Carcass Composition

Barley supplementation in Groups B and BL resulted in the production of higher amounts of meat, indicating improved muscle development. However, the total meat proportion did not differ significantly among the individual dietary treatments. The separable lean meat proportion was broadly within the range of 72.7–76% reported by Drew [45] for all farmed deer species younger than 26 months of age. Similarly, Volpelli et al. [11] reported higher weights of lean meat for FD fed concentrates in a daily amount of 500 g of dry matter/animal (40% maize, 25% sugar beet pulp, 20% alfalfa, 13% soy flakes, and 2% mineral and vitamins) compared with FD grazing on herbage for four months. However, the opposite trend was observed by Phillip et al. [12] in RD with slightly higher lean meat proportions in RD fed forage:concentrate at a 75:25 ratio than in those fed ratios of 50:50 and 25:75.

Due to the better growth performance, the supplemented deer in this study produced higher amounts of both high- and low-priced meat. However, the proportion of high-priced meat was higher in Group P, mainly due to the higher proportion of meat from the rump. Volpelli et al. [11] also reported a higher weight but a lower proportion of meat from the rump in FD supplemented with concentrates compared with those fed pasture.

The carcasses of Groups B and BL showed significantly higher amounts but lower proportions of bones and tendons. These results are consistent with those of feeding experiments comparing pasture-fed and concentrate-fed FD [11] and those of Phillip et al. [12], who reported decrease in bone yield proportion with increasing diet concentrate levels in RD.

In the present study, concentrate feeding in the form of barley had a major impact on carcass fatness, with the highest amount and proportion of separable fat in Group B. This increase in carcass fatness with grain-based diets is consistent with the results of Volpelli et al. [11] for FD and those of Phillip et al. [12] for RD. Similar to the effects on internal fat depots, the addition of dietary lysine decreased both the amount and proportion of separable fat in barley-supplemented deer. The decrease in carcass fatness might be a concern as venison is valued for its low but favorable fat content, and this result indicates that RPAA lysine supplementation could reduce the total fatness of FD meat/muscle. The effect of lysine on muscle fatness and, in turn, organoleptic quality warrants further research.

### 4.4. Physical Characteristics

As suggested by Hocquette et al. [46], the use of a single muscle as an indicator of the quality of all muscles in the carcass is not relevant. Therefore, in our study, we evaluated the effect of different dietary treatments on the physical characteristics of LL and SET muscles, which differ in anatomical location and function.

#### 4.4.1. pH Values

The pH of venison (defined as meat derived from cervids) after slaughter typically ranges from 6.5 to 7.2 and falls to a normal range of 5.8–5.3 within 24–48 h *post-mortem* [5,9,10,47]. However, due to various *ante-mortem* stress factors, such as transportation, chasing or the method of slaughter [18,48,49], FD are susceptible to the phenomenon known as dark, firm and dry (DFD) meat, which is typically found at pH_u_ > 6 [4]. In contrast to the higher pH_u_ values (pH_u_ > 6.0) reported for deer in Poland [4], which indicate an increased risk of DFD meat, no pH_u_ values > 6.0 were recorded in this work. In general, the differences in pH_u_ between the groups were small and not significant indicating that the *ante-mortem* stress was minimal and that the diets resulted in significant muscle glycogen reserves at point of slaughter. Previous studies also reported no effect of concentrate feeding on pH_u_ values in venison from FD [9,10,17] or RD [19].

#### 4.4.2. Meat Color

The color of meat strongly affects consumer acceptability and influences consumer purchase decisions [50]. As reviewed by Neethling et al. [51], consumers perceive bright, cherry-red meat as fresher and more wholesome than discolored and/or dark meat, which is usually associated with the DFD defect. However, as pointed out by Ramanzin et al. [52], a more intense or darker red color of venison seems to be acceptable to consumers because it is considered a typical feature of this type of meat. The typical darker and red-brown color of venison can be attributed to the higher content of myoglobin compared with meat of other domestic livestock species [53]. According to Volpelli et al. [10], this darker color is characterized by *L** value < 40, high *a** values and lower *b** values.

The meat from animals finished on pasture diets is usually darker than the meat from animals finished on concentrate diets [54]. Several factors are responsible for this difference, but pH and IMF appear to have the greatest impact. However, in this study, only minor differences in some color parameters were observed. Similarly, Volpelli et al. [10], Mulley et al. [55] and Hutchison et al. [9] reported no significant effect of diet with or without concentrates on the color parameters of muscles from FD. A possible explanation, as proposed in a review [52], is that commonly farmed ruminants have higher levels of (visible) IMF, which reflects light and thus gives a lighter appearance (higher *L** values). By contrast, cervids have low levels of IMF [12], even when fed supplements, and thus would not necessarily have statistically higher *L** values when supplemented compared with pasture/forage rearing only.

The *a** value of meat is closely related to myoglobin content, whereas the *b** value is related to the redness and lightness of meat. In general, the greater the myoglobin content in meat, the higher the values of *a** and *b** [56,57]. Significant differences were found between Groups P and B in the redness and yellowness of LL muscle. Since the *a** and *b** values are used to calculate chroma values, muscles with significantly higher *a** and/or *b** values will also have higher chroma values [47], as observed in the LL muscle of Group B. Similarly, Vestergaard et al. [57] reported higher redness, yellowness and chroma values in LL muscles of concentrate-fed bulls, although pigmentation was higher in bulls grazing pasture. Nevertheless, these color differences are probably primarily attributable to differences in physical activity rather than feeding level, as Group P likely spent more time foraging to consume sufficient feed, particularly during the dry period. This aspect of the effect of exercise on muscle color in FD warrants further evaluation.

Although not all of the detected differences were significant, the meat from supplemented FD appeared lighter, with more intense red and yellow color and higher chroma values than the meat from animals only grazing pasture. However, as reported by Hopkins and Nicholson [58], a direct effect of diet on meat color is considered rare and dependent instead on the effect of diet on muscle myoglobin and IMF. Nevertheless, the increases in the color of both muscles are consistent with the characteristics of venison that are considered attractive to consumers [12].

#### 4.4.3. Warner–Bratzler Shear Force

Meat tenderness is highly variable among animal species as a result of various extrinsic and intrinsic factors and these differences are mainly described by proteolytic enzyme activity, collagen content, muscle fiber characteristics and the anatomical position of each muscle [47]. While beef LL is classified into an intermediate tenderness category, the SET muscle is considered tough [59]. Cawthorn et al. [47] also noted differences in tenderness among FD muscles, with the LL muscle classified as inherently tender. Our results are in broad agreement with these observations, although the comparison of muscles was not the subject of this study.

As only minor differences in shear force values (meat tenderness) were observed in this study, it can be concluded that dietary treatment had no effect on the tenderness of FD meat. These results correspond with those reported for FD in Italy [10]. However, Mulley et al. [55] reported significantly lower shear force values for meat samples from FD fed concentrates compared with those grazing pasture. Similar results were obtained by Hutchison et al. [9] for FD in Australia. Supplementation with barley (800 g/animal/day) had an indirect effect on shear force values, and supplemented animals had a significantly higher IMF content and thus produced more tender meat than those with pasture fattening. It would be interesting to evaluate the effect of supplementary feeding on the chemical composition, particularly the IMF content, of FD fed supplementary barley during a very dry season like the one experienced in this investigation.

## 5. Conclusions

Given the increases in the popularity of deer farming and the accessibility of venison for the average consumer, the outcomes of this study are important for the assessment of the marketing quality of farmed FD. Supplementation with concentrates in the form of barley was an effective nutritional strategy to enhance the growth, performance and carcass yield of FD bucks in this study. This strategy led, however, to increases in internal and carcass fat content. Although the addition of RPAA lysine to the barley-supplemented diet decreased the total fatness of the FD, internal fat remained higher than that of FD reared on pasture alone. Further research is required to determine whether supplementation with lysine or other AAs could also improve other attributes of venison, which consumers consider a healthy alternative to traditional types of meat. By contrast, diet appeared to have a less pronounced effect on some of the physical parameters of LL muscle and no effect on the physical aspects of SET muscle. The pH_u_ values indicated no incidence of DFD meat, and color parameters were generally desirable for venison derived from FD. While effects of the different feed rations on the performance, carcass traits and physical attributes of FD meat were observed, these diets might also result in differences in chemical composition and organoleptic properties.

## Figures and Tables

**Table 1 animals-09-00033-t001:** Chemical composition of the pasture and barley supplemented to the animals.

Composition, g/kg Dry Matter	Barley	Pasture
Crude protein	11.27	12.74
Crude fat	2.44	1.91
Crude fiber	6.68	31.61
Ash	2.51	8.49
Nitrogen-free compounds	77.10	45.25
Lignin	0.83	5.00
Acid detergent fiber (ADF)	7.26	35.23
Neutral detergent fiber (NDF)	30.40	65.42

**Table 2 animals-09-00033-t002:** Growth performance and slaughter traits of fallow deer bucks (*n* = 45).

Item	Nutrition (Each *n* = 15)			SEM ^1^	*P*-Value
	Pasture (P)	Barley (B)	Lysine (BL)		
Initial weight (kg)	28.4	28.5	27.9	0.47	0.589
End weight—Phase I (kg)	37.1	39.1	38.2	0.56	0.054
Slaughter weight (kg)	45.3 ^B^	50.5 ^A^	49.8 ^A^	1.03	<0.001
Daily gain—Phase I (g/day)	97.1 ^B^	118.4 ^A^	115.0 ^A^	4.73	0.062
Daily gain—Phase II (g/day)	106.3 ^B^	147.0 ^A^	148.2 ^A^	9.28	<0.001
Daily gain—entire experiment (g/day)	100.7 ^B^	131.0 ^A^	130.1 ^A^	6.22	<0.001
Carcass weight (kg)	23.0 ^B^	28.4 ^A^	27.8 ^A^	0.48	<0.001
Dressing-out (g/kg)	508 ^B^	562 ^A^	559 ^A^	4.56	<0.001
Total internal fat (g/kg slaughter weight)	83 ^C^	197 ^A^	165 ^B^	7.81	<0.001

^1^ Standard error of the mean. ^A,B,C^ Within row, values with different superscripts differ significantly (*P* < 0.05).

**Table 3 animals-09-00033-t003:** Carcass composition of fallow deer bucks (*n* = 45).

Item	Nutrition (Each *n* = 15)			SEM ^1^	*P*-Value
	Pasture (P)	Barley (B)	Lysine (BL)		
**Weight (kg)**					
Right-side weight	11.26 ^B^	13.89 ^A^	13.65 ^A^	0.24	<0.001
Total meat	8.58 ^B^	10.63 ^A^	10.56 ^A^	0.20	<0.001
Bones and tendons	2.52 ^B^	2.84 ^A^	2.81 ^A^	0.05	<0.001
Separable fat	0.16 ^C^	0.42 ^A^	0.28 ^B^	0.04	<0.001
High-priced meat	5.24 ^B^	6.24 ^A^	6.29 ^A^	0.13	<0.001
Low-priced meat	3.34 ^B^	4.39 ^A^	4.27 ^A^	0.10	<0.001
**Right Side Proportion (g/kg Right-Side Weight)**					
Total meat	762.1	765.9	773.4	3.76	0.052
High-priced meat	465.2 ^A^	450.1 ^B^	462.2 ^AB^	3.36	0.004
Low-priced meat	296.9 ^B^	315.8 ^A^	311.2 ^A^	5.04	<0.001
Meat from: Rump	314.0 ^A^	298.5 ^B^	303.4 ^B^	2.12	<0.001
Shoulder	73.0 ^B^	73.2 ^B^	84.0 ^A^	2.57	<0.001
Loin	58.2	59.2	54.4	2.82	0.821
Tenderloin	20.0 ^AB^	19.2 ^B^	20.4 ^A^	0.54	0.033
Bones and tendons	223.4 ^A^	204.6 ^B^	205.7 ^B^	2.88	<0.001
Separable fat	14.5 ^C^	29.5 ^A^	20.9 ^B^	2.86	<0.001
**Ratio**					
Meat/bones	3.42 ^B^	3.75 ^A^	3.77 ^A^	0.06	<0.001
High/low-priced meat	1.57 ^A^	1.43 ^B^	1.48 ^B^	0.04	0.002

^1^ Standard error of the mean. ^A,B,C^ Within row, values with different superscripts differ significantly (*P* < 0.05).

**Table 4 animals-09-00033-t004:** Physical characteristics of *longissimus lumborum* and *semitendinosus* muscles from fallow deer bucks (*n* = 45).

Item	Nutrition (Each *n* = 15)			SEM ^1^	*P*-Value
	Pasture (P)	Barley (B)	Lysine (BL)		
***Longissimus lumborum***					
Weight (g)	833 ^B^	967 ^A^	942 ^A^	30.00	0.010
pH_u_	5.78	5.70	5.70	0.24	0.347
WB shear force (N)	24.63	26.59	26.14	2.16	0.295
**Color**					
Lightness, *L**	35.32	36.59	36.68	0.71	0.285
Redness, *a**	12.26 ^B^	14.98 ^A^	13.38 ^AB^	0.39	<0.001
Yellowness, *b**	9.90 ^B^	12.26 ^A^	11.31 ^AB^	0.44	0.002
Chroma	15.77 ^B^	19.39 ^A^	17.55 ^B^	0.52	<0.001
Hue angle	55.52	54.05	52.09	2.58	0.644
***Semitendinosus***					
Weight (g)	185	176	179	5.11	0.472
pH_u_	5.93	5.88	5.77	0.28	0.126
WB shear force (N)	41.77	39.21	43.37	2.41	0.252
**Color**					
Lightness, *L**	37.65	39.91	40.01	1.01	0.186
Redness, *a**	12.28	13.58	13.26	0.68	0.385
Yellowness, *b**	10.88	13.03	12.94	0.67	0.048
Chroma	16.47	18.88	18.60	0.86	0.110
Hue angle	48.01	40.44	39.66	3.88	0.255

^1^ Standard error of the mean. ^A,B,C^ Within row, values with different superscripts differ significantly (*P* < 0.05).

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
