# Peer review of "The Effect of Barley and Lysine Supplementation of Pasture-Based Diet on Growth, Carcass Composition and Physical Quality Attributes of Meat from Farmed Fallow Deer (Dama dama)"

_animals, 2019, doi:10.3390/ani9020033_

Round 1

Reviewer 1 Report

The manuscript is interesting scientific contributions to the knowledge of the effect of diet on growth, carcass composition and physical quality attributes of meat from farmed fallow deer (Dama dama). The paper has high scientific level, the experiment is well designed, the discussion is consistent and the final conclusions are interesting. Therefore, the manuscript may be published in Animals making major revision:

Suggestions for edition as well as some comments are the following:

Abstract

It´s Ok

Keywords

It´s Ok

Introduction

Please include one sentence about the effect of diet of fatty acid composition of deer meat

Line 93-95, please include this reference for this statement:

Lorenzo, J. M., Maggiolino, A., Gallego, L., Pateiro, M., Serrano, M. P., Domínguez, R., García, A., Landete-Castillejos, T., & De Palo, P. (2018). Effect of age on nutritional properties of Iberian wild red deer meat. Journal of the Science of Food and Agriculture, doi: 10.1002/jsfa.9334.

Material and methods

It´s Ok

Results and discussion

Authors need to include the results and discussion of the effect of diet on fatty acid composition of deer meat

Conclusions

It´s Ok

References

It´s Ok

Tables and figures

It´s Ok

I hope that my comments can improve the manuscript.

Author Response

Reviewer 1

Comments and Suggestions for Authors

The manuscript is interesting scientific contributions to the knowledge of the effect of diet on growth, carcass composition and physical quality attributes of meat from farmed fallow deer (Dama dama). The paper has high scientific level, the experiment is well designed, the discussion is consistent and the final conclusions are interesting. Therefore, the manuscript may be published in Animals making major revision:

Suggestions for edition as well as some comments are the following:

Abstract:

It´s Ok

Keywords:

It´s Ok

Introduction:

Please include one sentence about the effect of diet of fatty acid composition of deer meat

sentence about the effects of diet on fatty acids in deer meat has been added (Lines 58-61)

Line 93-95

Please include this reference for this statement:

Lorenzo, J. M., Maggiolino, A., Gallego, L., Pateiro, M., Serrano, M. P., Domínguez, R., García, A., Landete-Castillejos, T., & De Palo, P. (2018). Effect of age on nutritional properties of Iberian wild red deer meat. Journal of the Science of Food and Agriculture, doi: 10.1002/jsfa.9334.

as suggested, the above-mentioned article has been cited in the Introduction section of the manuscript  (Line 91) and in References (Lines 503-505)

Material and methods:

It´s Ok

Results and discussion

Authors need to include the results and discussion of the effect of diet on fatty acid composition of deer meat.

We definitely agree that the effect of diet on the fatty acid composition of deer adipose tissue might contribute to enhanced scientific knowledge. However, the present manuscript only deals with the effect of diet on growth, carcass composition and physical quality attributes of meat.

Conclusions:

It´s Ok

References:

It´s Ok

Tables and figures:

It´s Ok

Reviewer 2 Report

Dear authors,

Review of the article: “The Effect of Diet on Groth, Carcass Composition and Physical Quality Attributes of Meat from Farmed Fallow Deer (Dama dama)” by Kudrnáčová et al.

Title:

I recommend stating the title more precisely, e.g. “The Effect of Barley and Lysion Diet on Groth, Carcass Composition and Physical Quality Attributes of Meat from Farmed Fallow Deer (Dama dama)”. It emphasizes the importance of your article in comparison to others.

Simple Summary:

Line 26 ff.: Please delete the sentence “Moreover, papers dealing with...”. It is not necessary and the following sentence state the matter.

Line 27 ff.: Please delete the three words “To our knowledge”, these are not necessary too. Please add barley (“to evaluate the effects of dietary barley or barley and lysine supplementation...”.

Line 29: Please rewrite the whole sentence “Promising results”. You should not extol your results by words. Please be more precise in the general results.

Abstract:

Line 32: Please add Dama dama.

Line 43-46.: Please delete the sentences. They have no further information than the sentences before.

Keywords:

Line 47: I recommend changing “feeding system” into “diet”.

Introduction: No comments.

Material and methods:

Line 115: I recommend to change “L” into “BL” to underline the usage of barley.

Line 125: Please delete the space after “kg/day/”. Could you please write, if the percentage of lysine (5 g/day/animal LysiPEARL) changed in phase II of the diet?

Results and discussion:

Line 201 ff.: Please rewrite the sentence.

Table 2, 3, 4: I recommend to add (P), (B), (BL) in the table columns

Could you please state if there were any differences between the three slaughter charges (each 5 FD) of one dietary group.

Discussion:

Line 320 f.: Please delete the sentence.

Conclusions:

Line 400: Please change “and/or” into “or” (do the same in Line 61) and decide either to choose “improve” or “enhance”.

Line 407 f.: Please delete the last sentence.

Kind regards

Author Response

Reviewer 2 

Title:

I recommend stating the title more precisely, e.g. “The Effect of Barley and Lysine Diet on Growth, Carcass Composition and Physical Quality Attributes of Meat from Farmed Fallow Deer (Dama dama)”. It emphasizes the importance of your article in comparison to others.

 the title has been slightly modified (Lines 2-3)

Simple Summary:

Line 26 ff.

Please delete the sentence “Moreover, papers dealing with...”. It is not necessary and the following sentence state the matter.

sentence has been removed (Line 25)

Line 27 ff.

Please delete the three words “To our knowledge”, these are not necessary too. Please add barley (“to evaluate the effects of dietary barley or barley and lysine supplementation...”.

words “To our knowledge” have been removed (Line 25)

word “barley” has been inserted to the sentence as requested (Line 26)

Line 29

Please rewrite the whole sentence “Promising results”. You should not extol your results by words. Please be more precise in the general results.

the whole sentence has been re-written (Lines 27-30)

Abstract:

Line 32

Please add Dama dama.

Latin name Dama dama has been added  (Line 31)

Line 43-46

Please delete the sentences. They have no further information than the sentences before.

sentences have been removed (Lines 40-42)

Keywords:

Line 47

I recommend changing “feeding system” into “diet”.

 keyword has been changed (Line 43)

Introduction:

No comments.

Material and methods:

Line 115

I recommend to change “L” into “BL” to underline the usage of barley.

all “L” have been changed where appropriate (Lines 112, 120, 206, 211, 220, 224, 225, 227, 237, 254, 264, 266, 270, 284, 287, 294, 309)

Line 125

Please delete the space after “kg/day/”. Could you please write, if the percentage of lysine (5 g/day/animal LysiPEARL) changed in phase II of the diet?

space has been deleted (Line 122)

the sentence providing information about the dosage of lysine has been added (Lines 122-123)

Results and discussion:

Line 201 ff.

Please rewrite the sentence.

sentences have been re-phrased and paragraph organisation has slightly changed (Lines 198-206)

Table 2, 3, 4

I recommend to add (P), (B), (BL) in the table columns

(P), (B) and (BL) has been added to the table columns (Tables 2, 3, 4)

Could you please state if there were any differences between the three slaughter charges (each 5 FD) of one dietary group.

Day of slaughter was included as a random effect in the statistical model used in order to explain likely existing excess variability in dependent variables as we were not directly interested in quantifying the differences between different slaughter days.

Discussion:

Line 320 f.

Please delete the sentence.

sentence has been removed (Line 316)

Conclusions:

Line 400

Please change “and/or” into “or” (do the same in Line 61) and decide either to choose “improve” or “enhance”.

“and/or” has been changed into “or” (Lines 57 and 399) and “improve or enhance” in the Conslusions section has been modified to “improve” (Line 399)

Line 407 f.

Please delete the last sentence.

sentence has been removed (Line 406)

Round 2

Reviewer 1 Report

The authros must include the results of fatty acid